# Evaluation of the Properties of Eddy Current Sensors Based on Their Equivalent Parameters

**DOI:** 10.3390/s23063267

**Published:** 2023-03-20

**Authors:** Leszek Dziczkowski, Grzegorz Tytko

**Affiliations:** Faculty of Automatic Control, Electronics and Computer Science, Silesian University of Technology, Akademicka 16, 44-100 Gliwice, Poland

**Keywords:** eddy current testing, filamentary coil, I-core sensor, analytical modelling, sensor impedance

## Abstract

This paper presents a practical way of using the method of evaluating the metrological properties of eddy current sensors. The idea of the proposed approach consists of employing a mathematical model of an ideal filamentary coil to determine equivalent parameters of the sensor and sensitivity coefficients of tested physical quantities. These parameters were determined on the basis of the measured value of the real sensor’s impedance. The measurements were carried out with an air-core sensor and an I-core sensor while they were positioned at different distances from the surface of tested copper and bronze plates. An analysis of the influence of the coil’s position in relation to the I core on the equivalent parameters was also carried out, and the interpretation of the results obtained for various sensor configurations was presented in a graphical form. When equivalent parameters and sensitivity coefficients of examined physical quantities are known, it is possible to compare even very different sensors with the employment of one measure. The proposed approach makes it possible to make a significant simplification of the mechanisms of calibration of conductometers and defectoscopes, computer simulation of eddy current tests, creating the scale of a measuring device, and designing sensors.

## 1. Introduction

Eddy current testing is commonly used in many industries for the nondestructive evaluation of metal products. The tests are carried out using a single sensor in the form of a coil [1,2,3,4,5,6] or a system consisting of several coils constituting one sensor [7,8,9,10,11,12]. The basic idea of such an inspection is to utilise the phenomenon of electromagnetic induction. The alternating current flowing in the coil creates a magnetic field. If there is a conductive element within this field, eddy currents are induced in the conductor, creating a secondary magnetic field. According to Lentz’s rule, the secondary magnetic field is directed opposite to the primary field and thus causes a change in the impedance of the coil. This property makes it possible to detect defects in conductive objects [13,14,15,16,17,18,19] and determine their geometric dimensions [20,21,22,23,24,25], magnetic permeability [26,27,28], or electrical conductivity [29,30,31]. The changes in the impedance components of the sensor, and therefore the sensitivity of the measurement, strongly depends on the frequency of the current that excites the eddy currents and on the geometry of the sensor. In order to select the eddy current sensor construction that is optimal for a specific application, it is necessary to carry out many calculations and tests at the stage of its design. It is also important to properly select the frequency of the current exciting the coil since it affects not only the eddy currents penetration depth but also the metrological properties of the measuring instrument, including its sensitivity. Another important solution that needs to be implemented in the measuring instrument is the mechanism eliminating the influence of the lift-off effect [32,33] and the inaccuracy of the sensor position in relation to the tested element.

In all these applications, it is most convenient to employ the simplest possible coil model, which will significantly reduce the computational complexity and make it easier for the designer to select the optimal geometry of the sensor and test parameters. The mathematical model for a single-turn coil positioned over a conductive half-space was developed in [34,35]. The authors found in their research that, in many respects, it is much more convenient to use the model of an ideal filamentary coil (Figure 1). Such a coil contains *n* infinitely thin turns concentrated in a circle of radius *r*_0_, positioned at a distance *h*_0_ from the tested surface. Any coil used in eddy current testing can be associated with the corresponding equivalent parameters *r*_0_, *h*_0_ of a filamentary coil while the number of turns is maintained the same [36]. For this purpose, it is enough to determine such values of equivalent radius *r*_0_ and equivalent height *h*_0_ so that the impedance of the filamentary coil and the measured impedance of any real coil are equal. A mathematical model containing analytical expressions for the impedance of a filamentary coil placed over a two-layer conductive half-space [37] was derived using the truncated region eigenfunction expansion (TREE) method [38]. The algorithm in Matlab enabling the effective determination of the parameters *r*_0_, *h*_0_ was presented in [39] and subsequently used to detect defects with a filamentary coil [40]. The mathematical model for the system consisting of two coils was developed in [41,42,43,44,45,46]. In [47], it was proved that the change in the frequency and the change of the conductivity of the plate do not affect the change of the equivalent parameters. This convenient property was used by the authors while designing a sensor for a station for detecting cracks in clutches made with the method of powder metallurgy. The complicated shape of the tested elements necessitated the use of I-core sensors [48,49,50,51,52,53] with a small diameter. This core acted as a magnetic circuit and made it possible to move the coil away from the tested element. Such a thin core may touch the surface located in deep slots of the tested object.

In this work, the method of evaluating the properties of eddy current sensors with the employment of equivalent parameters *r*_0_, *h*_0_, described in [47], was used and extended. Section 2 presents the already-known analytical final formulas for the change in the impedance of the filamentary coil. Then, for the first time, a sensitivity analysis using a filamentary coil was performed for I-core sensors, which made it possible to determine the following:−the sensitivity of the resistance component to lift-off,−the sensitivity of the inductance component to lift-off,−the sensitivity of the resistance component to the electrical conductivity of the material under test,−the sensitivity of the inductance component to the electrical conductivity of the material under test,−the sensitivity of the resistance component to the magnetic permeability of the material under test,−the sensitivity of the inductance component to the magnetic permeability of the material under test.

The characteristics of the changes in sensitivity coefficients have been shown in the graphs. An algorithm for determining the sensitivity coefficients of the examined parameter in relation to changes in the resistance and reactance of the filamentary coil has also been presented. The Results section describes the measurements and calculations in detail. Equivalent parameters were determined for the air-core sensor and the I-core sensor. The analysis of what the changing of the distance of the sensor from the tested surface (lift-off) presented in [47] brings about was significantly extended. The novelty here is the detailed study of the influence of the coil position in relation to the core on the values of equivalent parameters, which were performed for various configurations. Subsequently, the obtained results were presented for 16 configurations in a graphical form, making it possible to observe the relationship between the geometry and position of the sensor and the value of equivalent parameters. In the Discussion section, the six most important conclusions for sensor constructors were formulated and discussed, thus enabling the practical application of the developed approach. A comparative analysis that describes how to correctly interpret the obtained values of equivalent parameters was also carried out.

## 2. Materials and Methods

The filamentary coil with *n* turns concentrated in a circle of radius *r*_0_ at a distance *h*_0_ from the conductive surface is shown in Figure 2. It was assumed that the thickness of the plate is greater than the depth of the eddy current penetration, which makes it possible to replace the plate in the mathematical model by means of half-space. The change in the coil impedance resulting from bringing it close to a material with conductivity *ϭ* and magnetic permeability *μ_r_* was formulated as follows [36]:(1)$$\Delta Z=j\omega \pi \;r_{0}^{}\mu_{0}\mu_{r}n^{2}\beta \int_{0}^{\infty }c(\lambda )\;e^{-\alpha \beta \lambda }J_{1}^{2}\left(\beta \lambda \right)d\lambda ,$$
where:(2)$$\alpha =\frac{2h_{0}}{r_{0}},$$
(3)$$\beta =r_{0}\sqrt{\omega \mu_{0}\mu_{r}\sigma },$$
(4)$$c(\lambda )=\frac{\lambda -\sqrt{\lambda^{2}+j}}{\lambda +\sqrt{\lambda^{2}+j}},$$
*ω* is the angular frequency, *μ*_0_ is the permeability of free space, and *J*_1_(*x*) is the Bessel function of the first kind of order 1.

When the impedance of the coil positioned at some distance from the conductor is denoted as *Z*_0_, and the impedance of the coil close to the conductor as *Z*, the change in impedance Δ*Z* = *Z* − *Z*_0_ can be represented as:(5)$$\Delta Z=R-R_{0}+j\omega (L–L_{0})=R-R_{0}-j\omega (L_{0}–L).$$

The changes in the impedance components were written in such a way so that both the change in resistance Δ*R = r = R* − *R*_0_ and the change in inductance Δ*L = l = L*_0_ − *L* were positive for non-ferromagnetic material. By determining the real part and the imaginary part of the impedance, the change in resistance *r* and the change in inductance *l* were subsequently calculated [36].
(6)$$r=R-R_{0}=n^{2}\omega \;\pi \mu_{0}\mu_{r}r_{0}\;\varphi (\alpha ,\beta ),$$
(7)$$l=L_{0}-L=n^{2}\pi \mu_{0}\;\mu_{r}r_{0}\;\chi (\alpha ,\beta ),$$
where:(8)φ(α,β)=Re{jβ∫0∞c(λ) e−αβλJ12(βλ)dλ},
(9)χ(α,β)=−Im{jβ∫0∞c(λ) e−αβλJ12(βλ)dλ}.

Using the exact differential equations, the Functions (8) and (9) can be written using the parameters *α* and *β*, defined in Equations (2) and (3).
(10)$$\Delta \varphi (\alpha ,\beta )=\frac{\partial \varphi }{\partial \alpha }\cdot \Delta \alpha +\frac{\partial \varphi }{\partial \beta }\cdot \Delta \beta ,$$
(11)$$\Delta \chi (\alpha ,\beta )=\frac{\partial \chi }{\partial \alpha }\cdot \Delta \alpha +\frac{\partial \chi }{\partial \beta }\cdot \Delta \beta ,$$
where:(12)$$\Delta \alpha =\frac{2}{r_{0}}\Delta h_{0},$$
(13)$$\Delta \beta =\frac{\beta }{2\sigma }\Delta \sigma .$$

According to Equations (6) and (7), the changes in the components of the coil depend on the changes in the functions *ϕ* (*α*,*β*) and *χ* (*α*,*β*). The changes in resistance and inductance of the coil can be written in the following form:(14)$$\Delta r=n^{2}\pi \frac{\beta^{2}}{r_{0}\sigma }\cdot \Delta \varphi ,$$
(15)$$\Delta \;l=n^{2}\pi \mu_{0}\mu_{r}r_{0}\cdot \Delta \chi .$$

Expressions (10)–(15) can be used in the sensitivity analysis to determine the influence of a given parameter on the change in the resistance and the inductance of a filamentary coil. The differentiation of Equations (6) and (7) brought about the determination of the sensitivity coefficients that define:−the change in the coil resistance resulting from changing the distance of its location from the tested surface (16),−the change in the coil resistance caused by changing the conductivity of the tested element (17),−the change in the coil inductance resulting from changing the distance of its location from the tested surface (18),−the change in the coil inductance caused by changing the conductivity of the tested element (19).
(16)$$\Delta rh_{0}=\frac{\Delta r}{\Delta h_{0}}=\frac{2n^{2}\pi \beta^{2}}{r_{0}^{2}\sigma }\frac{\partial \varphi }{\partial \alpha },$$
(17)$$\Delta r\sigma =\frac{\Delta r}{\Delta \sigma }=\frac{n^{2}\pi \beta^{3}}{2r_{0}\sigma^{2}}\frac{\partial \varphi }{\partial \beta },$$
(18)$$\Delta lh_{0}=\frac{\Delta l}{\Delta h_{0}}=2n^{2}\pi \mu_{0}\frac{\partial \chi }{\partial \alpha },$$
(19)$$\Delta l\sigma =\frac{\Delta l}{\Delta \sigma }=\frac{n^{2}\pi \mu_{0}r_{0}\beta }{2\sigma }\frac{\partial \chi }{\partial \beta }.$$

To denote the sensitivity, three-letter symbols have been adopted. The symbol Δ*r* means that the sensitivity is defined as the ratio of the change in resistance *r* to the change in a parameter of the tested element. Analogously, in the case of Δ*l*, it is the ratio of the change in inductance *l* to the change in a given parameter. The third letter of the sensitivity designation is the symbol of a parameter of the tested element.

Determining the sensitivity coefficients for changes in magnetic permeability is more complicated because *μ_r_* occurs not only in Equation (3) but also in Equations (6) and (7). At first, *μ_r_* was obtained from Equation (3) and substituted into Equations (6) and (7). In the next step, the differentiation of Equations (6) and (7) brought about the determination of the expressions for the change in resistance and inductance of the coil.
(20)$$\Delta r=[\frac{2\pi n^{2}\sqrt{\omega \mu_{0}\mu_{r}}}{\sqrt{\sigma }}\cdot \varphi (\alpha ,\beta )+\pi n^{2}r_{0}\;\omega \mu_{0}\mu_{r}\cdot \frac{\partial \varphi (\alpha ,\beta )}{\partial \beta }]\;\Delta \beta ,$$
(21)$$\Delta l=[\frac{2\;\pi n^{2}\sqrt{\mu_{0}\mu_{r}}}{\sqrt{\omega \;\sigma }}\cdot \chi (\alpha ,\beta )+\pi n^{2}r_{0}\;\mu_{0}\mu_{r}\frac{\partial \chi (\alpha ,\beta )}{\partial \beta }]\;\Delta \beta .$$

Then, the dependence Δ*β* = *f*(*μ*_r_) was determined from Equation (3).
(22)$$\Delta \beta =\frac{r_{0}}{2}\sqrt{\frac{\omega \;\sigma \mu_{0}}{\mu_{r}}}\Delta \mu_{r}.$$

In the final step, Equation (22) was substituted into Equations (20) and (21) and the sensitivity coefficients for magnetic permeability were obtained.
(23)$$\Delta \;r\mu_{r}=\frac{\Delta \;r}{\Delta \mu_{r}}=[\frac{2\;\pi \;n^{2}\sqrt{\omega \;\mu_{0}\;\mu_{r}}}{\sqrt{\sigma }}\;\varphi (\alpha ,\beta )+\pi \;n^{2}r_{0}\;\omega \;\mu_{0}\;\mu_{r}\;\frac{\partial \varphi (\alpha ,\beta )}{\partial \beta }]\;\frac{r_{0}}{2}\;\sqrt{\frac{\omega \;\sigma \;\mu_{0}}{\mu_{r}}},$$
(24)$$\Delta l\mu_{r}=\frac{\Delta \;l}{\Delta \mu_{r}}=[\frac{2\;\pi \;n^{2}\sqrt{\mu_{0}\;\mu_{r}}}{\sqrt{\omega \;\sigma }}\chi (\alpha ,\beta )+\pi \;n^{2}r_{0}\;\mu_{0}\;\mu_{r}\frac{\partial \chi (\alpha ,\beta )}{\partial \beta }]\;\frac{r_{0}}{2}\sqrt{\frac{\omega \;\sigma \;\mu_{0}}{\mu_{r}}}.$$

Expressions (23) and (24) can be written in a compact form:(25)$$\Delta \;r\mu_{r}=\pi n^{2}r_{0}\omega \mu_{0}\;\varphi (\alpha ,\beta )+\frac{\pi n^{2}r_{0}{}^{2}\sqrt{\omega^{3}\mu_{0}^{3}\mu_{r}\sigma }}{2}\cdot \frac{\partial \varphi (\alpha ,\beta )}{\partial \beta },$$
(26)$$\Delta l\mu_{r}=\pi n^{2}r_{0}\mu_{0}\chi (\alpha ,\beta )+\frac{\pi n^{2}r_{0}{}^{2}\sqrt{\omega \;\mu_{0}^{3}\;\mu_{r}\;\sigma }}{2}\cdot \frac{\partial \chi (\alpha ,\beta )}{\partial \beta }.$$

In order to present the characteristic of the changes in sensitivity coefficients, they were presented as a function of parameter *β* for several values of α (Figure 3). This choice results from the fact that the conductivity of the tested material in the expression for the change in the coil impedance (1) occurs only in the form of parameter *β*, and the distance *h*_0_ only in the form of parameter *α*.

Parameter *β_m_* is the maximum of function *ϕ* (*α,β*), and it determines the frequency for which the resistance of the sensor does not change along with the change in the conductivity. As distance *h*_0_ increases, all sensitivity coefficients (10)–(13) decrease, and the sensitivity of the instrument also decreases. The sensitivity coefficients Δ*rσ* and Δ*lσ* depend greatly and non-monotonically on the conductivity, frequency, and radius *r*_0_. This means that only the optimal selection of these parameters will ensure the best sensitivity of the sensor. Therefore, at the stage of designing a conductometer, flaw detector, and other eddy current devices, it is crucial to carefully select the geometry of the sensor and the frequency of the current. According to the authors, it is convenient to employ the method developed for this purpose, which consists in carrying out what follows:(1)In the first step, the components of impedance *Z*_0_ of the sensor positioned away from the conductive material should be measured. During the measurement, it is best to use the same frequency value as in the planned tests.(2)The second measurement is made for the same sensor after bringing it close to a conductive material of known conductivity *σ* and known magnetic permeability *µ_r_*. The impedance *Z* should be measured at the same frequency as in step 1.(3)On the basis of the measured impedance values, the changes in resistance Δ*R* = *R* − *R*_0_ and the changes in inductance *ΔL* = *L*_0_ − *L* caused by bringing the sensor close to the conductor are determined. In the case of the computer simulation of the tests or at the stage of designing the sensor, the changes in impedance components may be calculated with, for example, the finite element method or analytical expressions.(4)After obtaining the changes in the impedance components of the sensor, with the employment of the mathematical model of the filamentary coil, substitute parameters *r*_0_, *h*_0_ are calculated. At this stage, it is also possible to determine coefficients of the sensitivity of the analysed parameter to the change in the resistance and the change in the filamentary coil’s reactance.

When the values of *r*_0_, *h*_0_ and appropriate sensitivity coefficients are known, it is possible to precisely determine the properties of the sensor. The use of the same method of evaluation based on equivalent parameters for different sensors with different geometries makes it possible to compare them according to one criterion, which greatly simplifies the analysis process.

## 3. Results

In the measurements and calculations, a sensor in the form of a cylinder-shaped coil containing 700 turns was applied (Figure 4). The outer diameter of the coil was 25 mm, whereas the height was 14.8 mm. In the centre of the coil, there was a hole with a diameter of 8.1 mm, wherein an I core made of material F1001 with magnetic permeability *µ* = 3500 was placed. The employed core had a diameter of 8 mm and a height of 29.3 mm. The sensor impedance was measured using the Agilent E4980A precision LCR meter with an accuracy of ±0.1% for frequencies *f* = 10 kHz and *f* = 100 kHz. Each measurement was performed three times, and the arithmetic mean of the obtained values was calculated. In this way, the effect of random measurement error was reduced. Such an error is also influenced by the precision of the core installation and the positioning of the sensor in relation to the tested element. The influence of these parameters on the measurement result may be noticeable primarily for small values of the lift-off distance. In order to minimise this influence, washers were utilised, which ensured: the stabilisation of individual elements of the sensor and a position parallel in relation to the tested surface. The geometric dimensions of the coil and the core were measured using a micrometre screw gauge with an accuracy of ±0.05 mm.

In the first stage of the experiment, the impedance components of the air-core sensor positioned at different distances from the conductive surface, which were copper (*σ* = 58.42 MS/m) and bronze (*σ* = 10.47 MS/m) plates, were measured. The electrical conductivity of the plates was measured using a Foerster Sigmatest 2.069 instrument with an absolute accuracy of ±0.5% of the measured value. The thickness of the plates was greater than the depth of the eddy currents penetration. At the same time, the surface of each plate was large enough so that no edge effect was observed. In the next stage of the experiment, a core was placed in the coil. By changing the position of the coil in relation to the core and changing the distance of the sensor from the conductive surface, the impedance components of the sensor were measured. The measurements were carried out for 16 test configurations utilising the air-core sensor (Table 1) and for 32 configurations using the I-core sensor (Table 2). The lift-off distance of the sensor from the tested surface is marked as *h*. In the case of the sensor without a core, it is the distance from the lower edge of the coil to the surface of the tested element. If the sensor has a core, it is the distance from the core to the surface. In such a case, the distance from the bottom edge of the coil to the bottom edge of the core is *h*_c_. For each measurement, equivalent parameters *r*_0_, *h*_0_ of the filamentary coil were determined using the algorithm developed in [39]. The value of the filamentary coil impedance determined with the employment of the equivalent parameters was different from the measured value of the sensor impedance by less than 0.01%. The integration in Formula (1) was performed using Gauss–Kronrod quadrature implemented in Matlab. The absolute error tolerance of the integration in the numerical procedure was 1 × 10^−8^. The changes in frequency *f* and conductivity *σ* caused very small changes in the values of *r*_0_, *h*_0_—much smaller than the error of the method for measuring the impedance components. For this reason, Table 1 and Table 2 also show the average value of the equivalent parameters, which turned out to be very useful in the final assessment of the properties of the sensors.

The interpretation of the obtained results is shown in Figure 5, Figure 6, Figure 7 and Figure 8. Next to the dimensioned sketch of the sensor and the tested element, equivalent parameters *r*_0_, *h*_0_ are presented. Due to this, it is easy to compare the influence of positioning the coil in relation to the core and the influence of moving the sensor away from the tested surface on the value of the equivalent parameters and, thus, on the metrological properties of the sensors.
sensors-23-03267-t001_Table 1Table 1Equivalent parameters of the air-core sensor.








Average Value
*h*[mm]*h*_c_[mm]*f*[kHz]*σ*[MS/m]Δ*R*[Ω]Δ*X*[Ω]Δ*L*[mH]*r*_0_[mm]*h*_0_[mm]*r*_0_[mm]*h*_0_[mm]FigureNumber0-1058.425.8159.880.958.306.138.306.13Figure 5a and Figure 8a-10058.4220.07643.61.028.296.12-1010.4711.4451.250.828.316.14-10010.4744.87615.30.988.306.124-1058.421.6421.920.358.3710.128.3710.13Figure 5b and Figure 7a-10058.425.55231.50.378.3610.12-1010.473.3619.510.328.3710.14-10010.4712.56223.70.368.3710.128-1058.420.589.760.168.4214.248.4214.24Figure 5c-10058.421.94101.90.168.4214.23-1010.471.238.920.148.4314.25-10010.474.4499.20.168.4214.2316-1058.420.122.910.058.5022.458.5022.45Figure 5d-10058.420.4030.030.058.5022.45-1010.470.272.740.048.5022.46-10010.470.9329.470.058.5022.45
sensors-23-03267-t002_Table 2Table 2Equivalent parameters of the I-core sensor.








Average Value
*h*[mm]*h*_c_[mm]*f*[kHz]*σ*[MS/m]Δ*R*[Ω]Δ*X*[Ω]Δ*L*[mH]*r*_0_[mm]*h*_0_[mm]*r*_0_[mm]*h*_0_[mm]FigureNumber001058.4225.74362.65.7715.35.0815.305.12Figure 6a and Figure 8b10058.4284.6938126.0715.55.271010.4755.04325.15.1815.04.9910010.47195.336955.8815.45.14041058.4217.452734.3415.66.7115.686.76Figure 7b and Figure 8c10058.4257.1528544.5515.86.911010.4737.68247.63.9415.66.6410010.47123.227754.4215.76.78081058.4211.571963.1215.68.515.618.46Figure 8d10058.4237.820433.2515.88.711010.4725.16179.32.8615.68.2310010.4787.5619913.1715.58.39401058.4210.58155.92.4813.37.5513.357.56Figure 6b10058.4235.4616382.6113.47.561010.4722.1140.52.2413.47.5610010.4780.7615882.5313.47.56441058.427.79127.92.0414.09.2814.059.29Figure 7c10058.4225.9713362.1314.09.281010.4716.47116.51.8614.19.3110010.4759.3612992.0714.09.28801058.424.1173.791.1713.611.8913.6111.91Figure 6c10058.4213.68768.11.2213.611.881010.478.7467.841.0813.611.9810010.4731.30749.11.1913.611.87881058.422.4652.150.8314.615.5514.6315.53Figure 7d10058.428.13539.40.8614.615.511010.475.3348.610.7714.715.5410010.4718.7528.10.8414.615.531601058.420.9423.650.3814.421.1114.3721.11Figure 6d10058.423.09243.30.3914.421.081010.472.0722.30.3614.421.1510010.477.15238.90.3814.421.10


## 4. Discussion

The essence of the proposed approach lies in correctly interpreting the values of parameters *r*_0_, *h*_0_, which unambiguously characterise the metrological properties of eddy current sensors. Based on long-running studies and the results presented in this paper, the authors formulated and then discussed a number of conclusions that enable the effective and efficient application of the developed approach.

(1)The lower the value of *h*_0_, the greater the sensitivity of the sensor.

The intensity of the eddy currents induced in the tested material has a decisive influence on the sensitivity of the sensor. The smaller the distance *h*_0_, the closer the filamentary coil is to the conductor, so the intensity of the eddy currents increases (Figure 3). It is well known that the core concentrates the magnetic flux around itself, thus increasing the sensitivity of the sensor. How large this increase is can be assessed based on the value of parameter *h*_0_.

(2)The higher the value of parameter *r*_0_, the greater the changes in the impedance components caused by the change in the measured quantities.

Both in the process of designing the sensor and while selecting the optimal test parameters, it is important to obtain the highest value of *r*_0_ and, thus, the highest possible sensitivity of the sensor.

(3)The employment of the core brings “the coil magnetically closer to the tested material”.

The use of the core results in a decrease in the value of *h*_0_, which corresponds to bringing the coil in contact with the conductor. Parameter *h*_0_ may be treated as a measure of the efficiency of the core performance.

(4)The higher the value of parameter *r*_0_, the larger the area of the tested element where eddy currents that change the components of the coil impedance are induced.

Parameter *r*_0_ depends primarily on the inner radius and the outer radius of the real coil. Along with an increase in the value of these radii, the coverage of eddy currents induction in the conductor also increases. This property should be utilised appropriately according to the application of the sensor. In the case of testing large surfaces, such as plates or metal sheets, designers aim to obtain the most extensive inspection coverage. However, there are applications, such as rod testing, coin sorting or crack detection in small-diameter cylinders, where a short inspection coverage is preferable. This requirement results from the fact that various factors disturbing the measurement occur in close proximity or that there is the likelihood of an edge effect.

(5)The higher the value of *h*_c_, the lower the sensitivity of the sensor.

When height *h*_c_ increases due to moving the coil “up” along the core, the value of equivalent parameters increases. The sensitivity of the sensor decreases because a small increase in *r*_0_ cannot compensate for a very large increase in *h*_0_.

(6)The number of turns of the coil does not affect the value of equivalent parameters.

Changing the number of coil turns does not change the value of *r*_0_, *h*_0_. The parameter specifying the number of turns appears as *n*^2^ both in the formula for the impedance of the real coil and in the formula for the filamentary coil (1). This property constitutes an important advantage over the analysis with a single-turn coil. Only in the case of a filamentary coil is it possible that the number of turns does not impact the results of the comparison and evaluation of the properties of eddy current sensors.

The above conclusions were used to perform an exemplary comparative analysis. The equivalent parameters determined for the air-core sensor placed on the tested surface are *r*_0_ = 8.3 mm and *h*_0_ = 6.13 mm. After having inserted the core, the value of parameter *h*_0_ decreases (*h*_0_ = 5.12 mm), thus increasing the sensitivity of the sensor. At the same time, parameter *r*_0_ increases from 8.3 mm to 15.3 mm, which causes an additional increase in sensitivity due to the increase in the magnetic field range. Similarly, it is possible to analyse the effect of moving the sensor away from the tested surface. The air-core sensor that is brought into contact with the tested surface has an equivalent height of *h*_0_ = 6.13 mm. After moving the sensor 4 mm away from the tested surface, *h*_0_ = 10.13 mm, i.e., Δ*h* = 10.13 − 6.13 = 4 mm. If the sensor is 8 mm away from the tested surface, *h*_0_ = 14.2 mm, i.e., Δ*h* = 14.2 − 6.13 = 8.07 mm. Moving the air-core sensor away from the surface of the tested element by a particular value increases *h*_0_ by the same value. In the case of the I-core sensor, the changes in parameter *h*_0_ are slightly different. Such a sensor, when brought into contact with the tested surface, takes the value of parameter *h*_0_ = 5.12 mm. When the sensor is moved 4 mm away from the tested surface, *h*_0_ = 7.56 mm, i.e., Δ*h* = 7.56 − 5.12 = 2.44 mm. Moving the sensor away from the tested surface by 8 mm brings about an increase of the value of parameter *h*_0_ = 11.91 mm, i.e., Δ*h* = 11.91 − 5.12 = 6.79 mm. The beneficial effect of the core is apparent. For small distances, moving the sensor away increases the equivalent height *h*_0_, however, by a value that is smaller than the actual distance. Thus, the sensitivity of the I-core sensor decreases less significantly than that of the air-core sensor.

## 5. Conclusions

This paper presents a practical way of using the method of evaluating the metrological properties of eddy current sensors with the use of equivalent parameters of a filamentary coil. The presented sensitivity analysis makes it possible to obtain expressions that define the influence of a selected parameter of the sensor-tested element system on changes in the coil impedance components. The developed geometric interpretation of changes in sensitivity coefficients allows a better understanding of the characteristic of changes in the analysed sensitivity coefficients. It is the first time that the question of how the position of the coil in relation to the core affects the equivalent parameters has been examined. In the measurements, a sensor with a removable I-core was used. Due to this, it was possible to observe the influence of the core on the metrological parameters of the sensor. The measurements were carried out for different frequency values using thick copper and bronze plates. The sensor was positioned at different distances from the tested surface, while the position of the coil with respect to the core was also changed. The measured values of the impedance changes were employed to determine the equivalent parameters. The results of the calculations confirmed that parameters *r*_0_, *h*_0_ may be successfully used to evaluate the properties of eddy current sensors.

The presented method for describing the sensors using two numbers that are equivalent parameters of *r*_0_, *h*_0_ is easy to perform and provides a number of benefits. The usefulness of the proposed concept consists in enabling the comparison of very different sensors with the employment of the same measure. It is enough to determine the sensitivity coefficients of the examined physical quantities with the application of the universal filamentary coil model. Then, equivalent parameters, which are the dimensions of the ideal filamentary coil, are determined for the real sensors. On the basis of the once-determined values of *r*_0_, *h*_0_ and sensitivity coefficients, the metrological properties of the sensors can be assessed. The proposed method can be utilised for many applications, such as the designing of sensors, selection of test parameters, computer simulations, interpretation of the obtained results, etc.

In this article, the properties of I-core sensors were analysed due to the fact that they were used in the designed device. In further work, sensors with different geometry will also be employed to detect various types of defects. In addition, tests of conductors of other shapes, including roughness and porosity of the tested surface, are being planned.

## Figures and Tables

**Figure 1 sensors-23-03267-f001:**
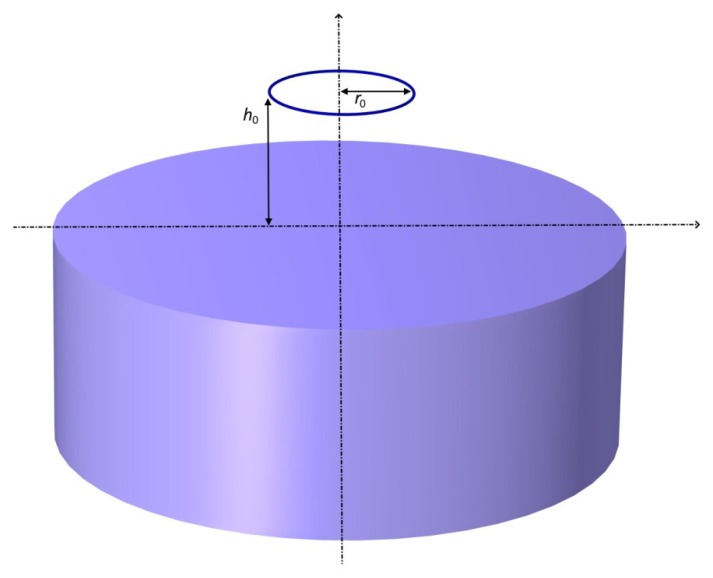
Filamentary coil placed over a conductive cylinder.

**Figure 2 sensors-23-03267-f002:**
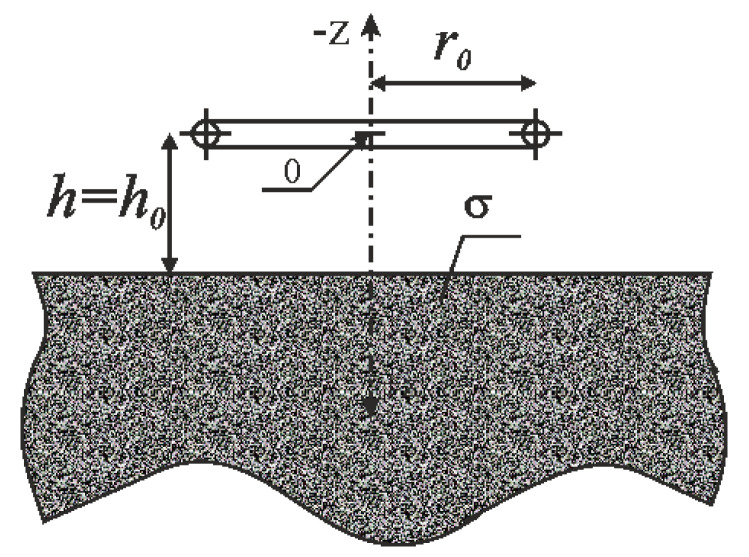
Rectangular cross–section of the filamentary coil located above the conductive half–space.

**Figure 3 sensors-23-03267-f003:**
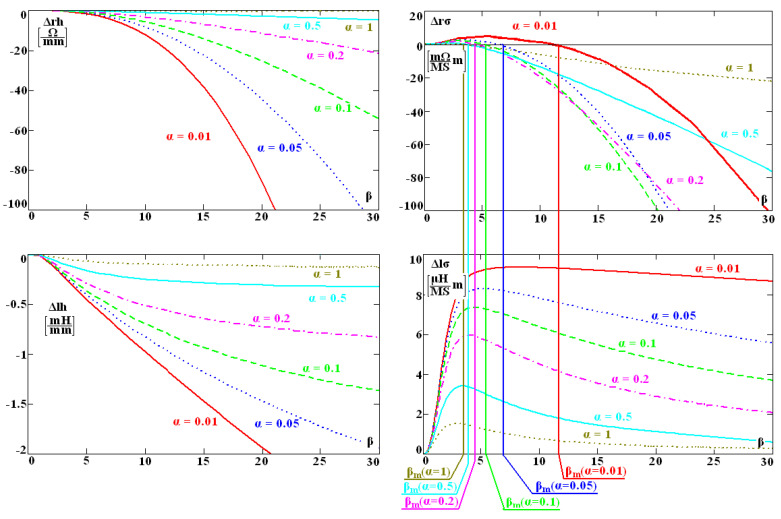
Sensitivity of resistance and inductance of the filamentary coil to conductance variations of the examined workpiece material as well as to variations of the distance between the measuring coil and the examined surface. The graphs are plotted as functions of the generalised *β* parameter and for several values of *α*.

**Figure 4 sensors-23-03267-f004:**
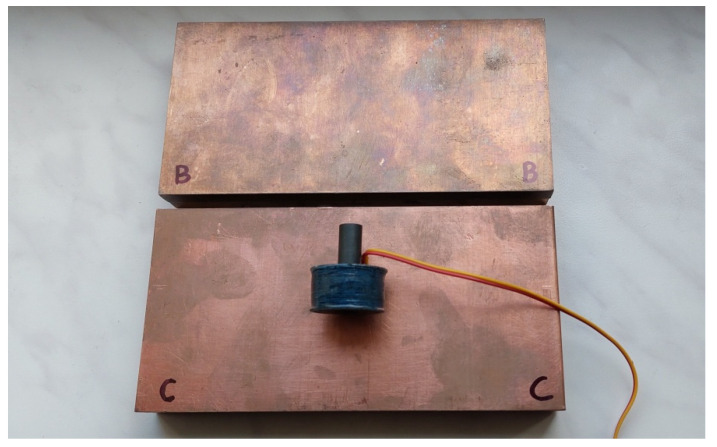
I-core sensor and plates made of bronze (B) and copper (C) used in measurements.

**Figure 5 sensors-23-03267-f005:**
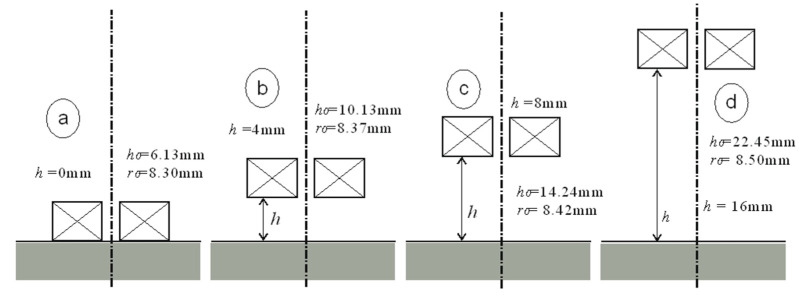
Equivalent parameters of the air-core sensor for height *h*: (**a**) *h* = 0 mm, (**b**) *h* = 4 mm, (**c**) *h* = 8 mm, (**d**) *h* = 16 mm.

**Figure 6 sensors-23-03267-f006:**
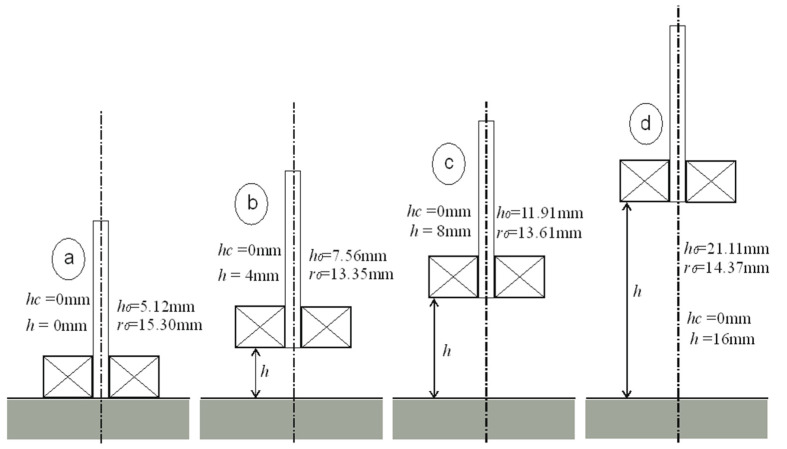
Equivalent parameters of the I-core sensor for height *h*_c_ = 0 mm: (**a**) *h* = 0 mm, (**b**) *h* = 4 mm, (**c**) *h* = 8 mm, (**d**) *h* = 16 mm.

**Figure 7 sensors-23-03267-f007:**
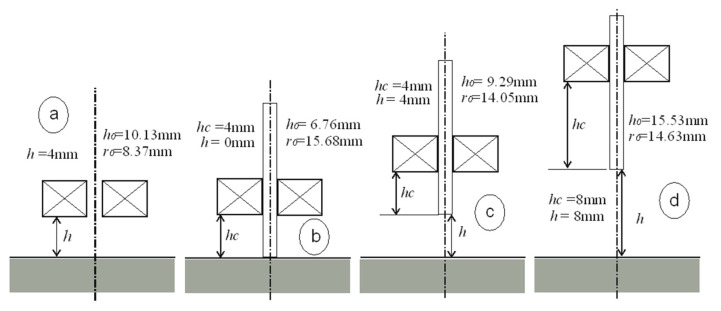
Equivalent parameters of the air-core sensor and the I-core sensor: (**a**) *h* = 4 mm, (**b**) *h* = 0 mm and *h*_c_ = 4 mm, (**c**) *h* = 4 mm and *h*_c_ = 4 mm, (**d**) *h* = 8 mm and *h*_c_ = 8 mm.

**Figure 8 sensors-23-03267-f008:**
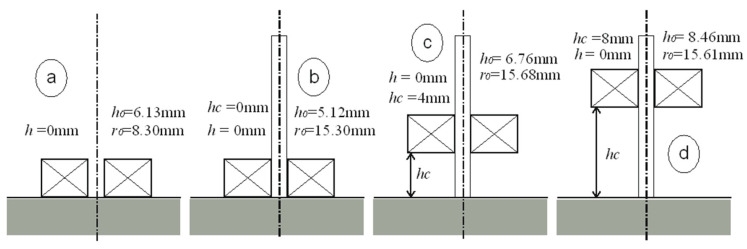
Equivalent parameters of the air-core sensor and the I-core sensor, for height *h* = 0 mm: (**a**) without the core, (**b**) *h*_c_ = 0 mm, (**c**) *h*_c_ = 4 mm, (**d**) *h*_c_ = 8 mm.

## Data Availability

Not applicable.

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
