# Peer review of "Evaluation of the Properties of Eddy Current Sensors Based on Their Equivalent Parameters"

_sensors, 2023, doi:10.3390/s23063267_

Round 1

Reviewer 1 Report

The paper relies on expressing the impedance of a circular filament coil. That approach is not particularly new, as most of the eddy current sensor modeling approaches incorporate that approach one way or the other with slight variations. Therefore, the authors must emphasize better what is exactly new in their approach and what value the paper adds. In that light, the sentence "In this work, it is the first time that the equivalent parameters r0, h0 of the filamentary coil have been used to evaluate the properties of eddy current sensors." -- becomes a highly disputable statement. It is a very strong claim the authors make, and I doubt that claim is true. If it is true, the authors must present a very strong argument as to why that is true. Authors must describe what has been done in terms of using r0, h0 before, and what exactly is new about the authors' work. 

In the equations (1) to (4), some variables are not defined. Make sure all variables are clearly defined. That shortcoming is carried forward to other equations also. 

Also, cite references for each equation where they were taken from previous works. Then the authors must try to emphasize what is new in terms of equations. Are there any original derivations? Authors must emphasize which part is original, and which equations are taken from previous works. It helps when each equation carries a reference if they were taken from previous works.

Improve figure 1 to show r0 and h0.

Reviewer 2 Report

The paper describes a method for evaluating the sensitivities (electrical resistance and inductance vs geometry and material conductance) of eddy current sensors employing a mathematical model of an ideal filamentary coil.

The method seems to be not innovative being that it has already been published by authors in reference [47], a work in which the same approach, similar experimental results, and comments are provided. If the understanding is wrong, please clarify the innovative content with respect to the reference either in the text, or summarize/remove the theoretical parts already developed or identified in previous works.

The method wants to find two equivalent parameters, r0 and h0 to identify the sensitivity of the eddy current sensors, but it is not clear (and quantified) the measurement uncertainty of the proposed approach. The uncertainty of the method depends on the uncertainty of the measurements on which is based the extraction of r0 and h0 and on the numerical optimization (if more than one measurement is performed for each condition), an evaluation which is not presented in the proposed work.

Moreover, in order to assess the robustness of the proposed method and evaluate any deviation from the calibration, it is necessary to compare the prediction of the indirectly measured sensitivities with the ones obtained by typical calibration, being that effects like local non-linearities or hysteresis (general output from performed calibration) cannot be identified at this stage of the work.

Finally, in the discussion, I would suggest avoiding qualitative expressions like  “it is vital” and quantifying the claimed achievements.

Round 2

Reviewer 1 Report

The authors are still not getting the message out elegantly.

The descriptions must improve.

Do not confuse the Introduction with the phrase "equivalent parameters". If using that phrase, use it with care because at one point the authors refer to r0 and h0 as equivalent parameters, then at another point authors say "Equivalent parameters were determined for the air-core sensor and the I-core sensor using different values of frequency and electrical conductivity." That does not sound right as r0 and h0 are geometric parameters and do not depend on frequency. In the latter sentence the authors are discussing another set of parameters. Thus the message is convoluted and is not coming out as clearly and elegantly as it should. 

If I was writing this paper, I would present the last half of the Introduction as follows:

I would draw a clear diagram depicting the sensor geometry and its placement -- something like those in Figure 5 and beyond. After indicating the realistic geometry of the sensor, I will describe the "equivalent geometric parameters" r0 and h0. I might even draw them in the same diagram on top of the "real geometry" of the sensor. Doing so is essential to make it possible for someone else to repeat similar experiments.

Also, it is important at this point to specify about the material under test. In the authors equations in Section 2 they indicate that the material has a relative permeability -- meaning that it is magnetic. But in the further experiments I do not see the sensitivity analysis extended to permeability. Why? Is the material magnetic or non-magnetic? Please specify. If it is magnetic, why is the sensitivity analysis not extended to permeability? Please specify. 

Then I would describe that I am modeling the sensor's impedance. Say that the impedance is modeled as a resistance component and an inductance component. Then, say that the objective is to present a four-fold sensitivity analysis -- which is the contribution of the paper. The four fold sensitivity analysis can be summarized here as: (1) The sensitivity of the resistance component to lift-off; (2) The sensitivity of the inductance component to lift-off; (3) The sensitivity of the resistance component to the electrical conductivity of the material under test; (4) The sensitivity of the inductance component to electrical conductivity of the material under test.

As said before, a question naturally arises about sensitivity to magnetic permeability as well, since the sensitivity to electrical conductivity is studied. Therefore, the authors must justify the case as to why the sensitivity to permeability is not studied.

An Introduction as such would be clearer. I have seen the authors have added a new paragraph after the first review. Amending that same paragraph to reflect the above points would suffice.

Section 2: Mathematical formulation. It seems the authors are repeating previous work. No original formulations it sounds to me. That is fine. But, in that case, the authors must give references to each equation to avoid confusion. For instance, in the current draft, no reference is mentioned for equations (10) to (13). If they are original derivations, the authors must declare so, or, if they are extracted from elsewhere, the references must be given.

Also, instead of repeating just text book equations (from [36] -- a reference from 1971) -- can the authors make the equations more complete and self-sufficed? For example, can the authors express the partial derivative terms in the right hand sides of equations (10) to (13)? Expressing them also, will make the formulation much more complete.

Then after that, the authors presenting Figure 3 -- the graphs -- is confusing. Need a lot more detail before getting to those graphs. Section 2 will stand alone with a rich description of formulas. The graphs will sit better as results. But make sure no gaps are left--i.e., all the details, including complete equations and parameter values tested, must be given before plotting the graphs.

Authors are advised to shuffle the paper more to get the message out more compellingly and clearly.      

Reviewer 2 Report

Dear Authors, thank you for the clarifications/modifications.

Author Response

Dear Professor,

Thank you very much for your review.